# Enhancing Dynamic Bandwidth of Amplified Piezoelectric Actuators by a Hybrid Lever and Bridge-Type Compliant Mechanism

**Mingxiang Ling [1,2], Lei Yuan [1]****, Zhihong Luo [3], Tao Huang [3,*] and Xianmin Zhang [1,*]**

[1]  School of Mechanical and Automotive Engineering, South China University of Technology, Guangzhou 510641, China; ling_mx@163.com (M.L.); yl18374459889@163.com (L.Y.)
[2]  Institute of Systems Engineering, China Academy of Engineering Physics, Mianyang 621999, China
[3]  The College of Mechanical Engineering, Chongqing University, Chongqing 400044, China; 202007021012@cqu.edu.cn
[*]  Correspondence: thuang@cqu.edu.cn (T.H.); zhangxm@scut.edu.cn (X.Z.)

**Abstract:** Ongoing interests in high-speed precision actuation continuously sparks great attention on developing fast amplified piezoelectric actuators (APAs) with compliant mechanisms. A new type of APA with enhanced resonance frequency is herein reported based on a hybrid compliant amplifying mechanism. A two-stage displacement flexure amplifier is proposed by synthesizing the lever-type and semi bridge-type compliant mechanisms in a compact configuration, promising to a well tradeoff between the displacement amplification ratio and dynamic bandwidth. The static and dynamic performances are experimentally evaluated. The resonance frequency of 2.1 kHz, displacement amplification ratio of 6, and step response time of around 0.4 ms are realized with a compact size of 50 mm × 44 mm × 7 mm. Another contribution of this paper is to develop a comprehensive two-port dynamic stiffness model to predict the static and dynamic behaviors of the compliant amplifier. The modeling approach presented here differs from previous studies in that it enables the traditional transfer matrix method to formulate both the kinetostatics and dynamics of compliant mechanisms including serial-parallel branches and rigid bodies.

**Keywords:** compliant mechanisms; piezoelectric actuator; displacement amplifier; flexure hinges; transfer matrix method





## 1. Introduction

Piezoelectric actuators consisting of a stack of piezoceramic layers electrically connected in parallel are characterized by fast dynamic response, nano-scale resolution, and large blocking force. However, the stroke of many commercially available piezoelectric stacks is limited to dozens of microns with the strain of around 0.1%. Such a drawback should be addressed since plenty of precision actuation applications, including positioning [1–3], gripping [4–6], scanning [7], manufacturing [8], switching [9], manipulating [10], etc., often require sub-millimeter or even larger strokes. Therefore, much research has been devoted to mechanically amplifying the micro stroke of piezoelectric stacks [11–13]. Such a technique is usually termed as 'amplified piezoelectric actuators' (APAs) [14], as exemplarily illustrated in Figure 1.

Recently, more and more requirements toward the high actuation speed have come of age, such as scanning probe microscopy, high-throughout micro/nano manufacturing, high-speed jet dispensing, and fast mechanical switches [15,16]. The motion rate of amplified piezoelectric actuators is inherently restricted by their resonance frequencies, and the dynamic response bandwidth of amplified piezoelectric actuators will generally be reduced as their travel range is amplified. The amplitude of motion will become larger and larger with the increase of working frequencies until reaching the fundamental resonance mode,

then the dynamic response will be attenuated and become uncontrollable at the domain of high vibration modes [17]. Moreover, when inadvertently excited, input electric signals with high-frequency components will excite the natural motion oscillation, causing the loss of precision and stability. Thereby, limiting the operating frequencies to below one third of the lowest vibration mode or even smaller scopes are common in actual engineering applications. Feedback and feedforward control strategies [18,19], such as the positive position feedback, integral resonant, and shunt controllers, have been proposed to dampen vibration modes enabling higher actuation rates. Nevertheless, inherently increasing the mechanical bandwidth from the viewpoint of mechanism design is a primary concern. It involves careful considerations of the mechanical property of flexure hinges, the synthesis of optimal configurations as well as the interfacial treatment between piezoelectric stacks and compliant mechanisms.

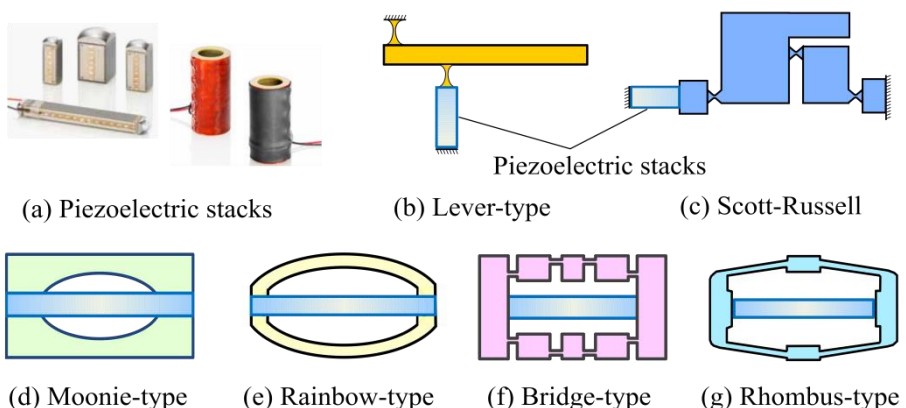

**Figure 1.** Picture of piezoelectric stacks and schematic diagram of common types of mechanically amplified piezoelectric actuators by compliant mechanisms.

A number of designs for amplifying the micro displacement of various actuators have been demonstrated successfully in the literature, these include the early rainbow and Moonie-type flextensional mechanisms [20], the now mainstream bridge-type [21,22], lever-type [23], Scott-Russell [24] complaint mechanisms, and more recently composite amplifying mechanisms with configurations of nested cellular and multi-stage ones [25–27], to name a few. These techniques, as a whole, can successfully amplify the micro stroke of piezoelectric stacks and other actuators with varied load capacity and dynamic bandwidth in different sizes. Many investigations also reported designs of hybrid compliant amplifying mechanisms to realize a large displacement amplification ratio by combining the bridge-type and leveraged mechanisms [28,29], Scott-Russell and bridge-type mechanisms [30,31], Scott-Russell and leveraged mechanisms [32,33], and other types of hybrid compliant mechanisms [34]. For example, Kim et al. [35] proposed a three-dimensional amplified piezoelectric actuator based on a two-stage bridge-type compliant mechanism with the output displacement of 80 μm and the resonance frequency of 190 Hz. By contrast Ding et al. [31] combined bridge-type and Scott-Russell compliant mechanisms to amplify the micro stroke of piezoelectric actuator with the output displacement of 69 μm and the resonance frequency of 457 Hz. Some other designs of amplified piezoelectric actuators can be found in literature such as those with the output displacement and resonance frequency of 30 μm and 1152 Hz in [7] as well as 200 μm and 189 Hz in [36], to name a few. As discussed, pursuing a higher displacement amplification efficiency with a compact configuration and importantly enhancing the dynamic bandwidth is still a challenging issue. Attempts on equilibrating such a tradeoff are attractive for high-speed and small-space engineering applications.

The design presented in the current study differs in that it optimally configure the lever-type and semi bridge-type compliant mechanisms to design a hybrid two-stage amplified piezoelectric actuator with enhanced dynamic bandwidth and displacement

amplification efficiency in a much compact configuration. Another contribution of this paper is to predict the static and dynamic performances of the flexure amplifier with a comprehensive two-port dynamic stiffness model. The modeling approach presented here differs from the previous ones in that it enables the traditional transfer matrix method to formulate both the kinetostatics and dynamics of compliant mechanisms with complex branches and rigid bodies in serial-parallel configurations.

The rest of this paper is organized as follows: The structural design is introduced in Section 2. Comprehensive two-port dynamic stiffness modeling is carried out in Section 3, followed by parameter influence analysis in Section 4. The prototype is experimentally tested in Section 5. This paper concludes in Section 6 with a summary of results.

## 2. Operational Principle and Configuration

As has already been demonstrated in [37], a large ratio of output stiffness of the former stage to the input stiffness of the next stage in multi-stage compliant amplifying mechanisms will ensure a large composite displacement amplification ratio. This indicates that synthesizing a flexure amplifying modular having high output stiffness with the one having low input stiffness is beneficial to enhance the achievable output displacement of a multi-stage compliant amplifying mechanism. Based on the above, we combine the lever-type and semi bridge-type compliant amplifying mechanism to develop a two-stage displacement amplification mechanism, as schematically illustrated in Figure 2. The output ports of two mirror-symmetric levers are rigidly connected to the input ports of the semi bridge-type amplifier. V-type flexure hinges with the merits of high bending stiffness and small parametric motion errors are adopted to enhance the output stiffness of the leveraged amplifier at the first amplifying stage. Micro stroke of the internal piezoelectric actuator is mechanically amplified twice, serially by the lever principle and flextensional effect with the loss of a certain level of blocking forces.

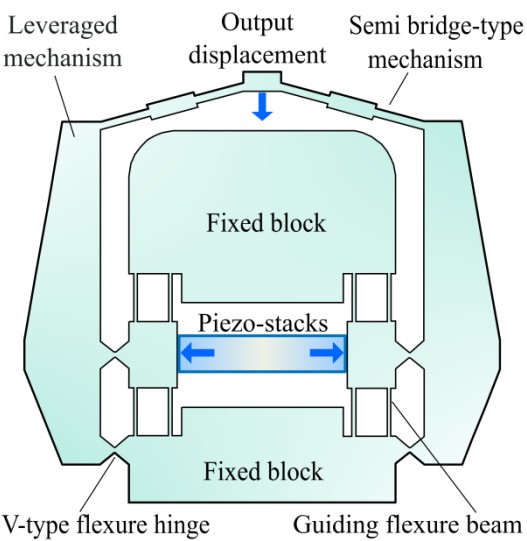

**Figure 2.** Schematic diagram of the presented hybrid two-stage displacement amplification mechanism by synthesizing lever-type and semi bridge-type compliant mechanisms.

The key features of our design lie in its compact configuration and minimized parasitic motion errors due to the monolithic and symmetric structure. It is also easy to integrate a displacement sensor without the requirement of additional spaces, as clearly shown in Figure 3. The design of embedded displacement sensor also contributes to the miniaturization of the whole application system. In addition, using flexure modules as few as possible is helpful to keep a high transfer efficiency of electric–mechanical energy from piezoelectric stacks to the output port of the compliant amplifying mechanism. The loss of blocking force can be relieved to a certain level with a smaller number of used flexure

modules. Importantly, the inertial motion of piezoelectric stacks is avoided in the presented design in contrast to the traditional bridge-type compliant amplifiers [21,22]. Based on these concerns, a refined displacement amplification efficiency and enhanced dynamic bandwidth with a compact size can be expected.

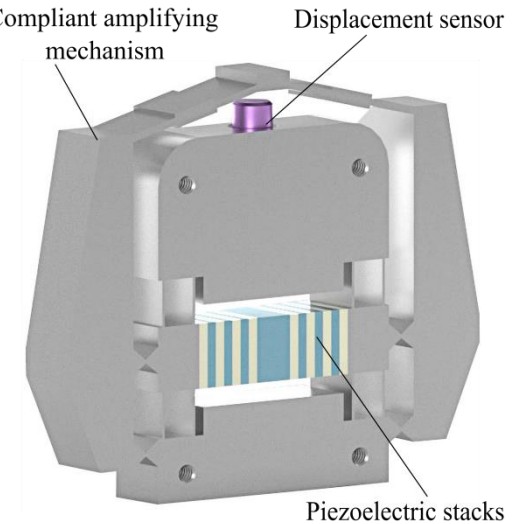

**Figure 3.** Three-dimensional view of the presented hybrid two-stage displacement amplification mechanism with assembled piezoelectric stacks and potential displacement sensor.

Two groups of guiding flexure beams are designed on the input port of the levers. Based on the simulated deformation nephogram in Figure 4, the parasitic rotational motion of the two input ports are avoided by designing guiding flexure beams, which protects piezoelectric stacks from lateral and shear forces. More importantly, the parasitic rotational motion of the two input ports without guiding flexure beams will reduce the contact stiffness between the compliant mechanism and piezoelectric stacks, thus attenuating the overall output displacement. It can be clearly seen from Figure 4b that the input ports are able to translationally move along the horizontal direction without obvious parasitic rotational motions after adding two pairs of guiding flexure beams. Furthermore, assembling piezoelectric stacks into the compliant mechanism with an interference fit and preload will ensure a well interfacial contact.

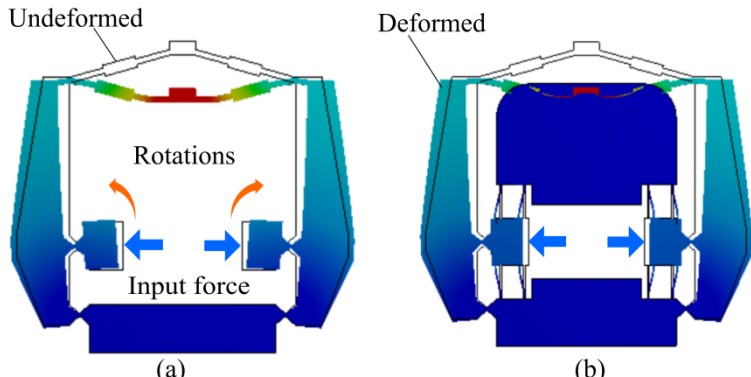

**Figure 4.** Parasitic motion errors with and without guiding flexure beams simulated by ANSYS software package. (**a**) Without the guiding flexure beams. (**b**) With the guiding flexure beams. It is noticed that the finite element simulation presented here is only for illustrating the parasitic motion errors. The detailed settings for the numerical simulation can be found in Section 4.

## 3. Parametric Formulation

In this section, a comprehensive two-port dynamic stiffness model is derived to parametrically characterize the kinetostatics and dynamics of the displacement amplification mechanism. The two-port dynamic stiffness model captures the linear kinetostatics and dynamics of a compliant mechanism from the viewpoint of input and output ports with a pseudo-static characteristic on the frequency domain. In contrast to our previous study [38], a comprehensive two-port dynamic stiffness model is developed here, with which the traditional transfer matrix method is improved to be able to formulate serial-parallel flexure-hinge mechanisms including rigid bodies.

The amplifier is discretized into flexure hinges/beams, rigid bodies, and lumped mass shown in Figure 5. The input and output ports are concentrated as lumped masses. The levers are denoted as rigid bodies whose rigid motion should be considered. The flexure hinges/beams are numbered serially from elements (1) to (14) and connected with nodes labeled as a circle in Figure 5. During the numbering, the end node of flexure hinges/beams connected to a rigid body were shifted to the mass center of rigid body by including the kinematic effect [39]. Input force and displacement from piezoelectric stacks are denoted as $f_{in}$ and $x_{in}$. A dummy force $f_{out}$ is exerted on the output port, and there has: $f_{in2} = -f_{in1} = [f_{in}; 0; 0]$, $x_{in2} = -x_{in1} = [x_{in}; 0; 0]$, $f_{out} = [0; f_{out}; 0]$, $x_{out} = [0; x_{out}; 0]$.

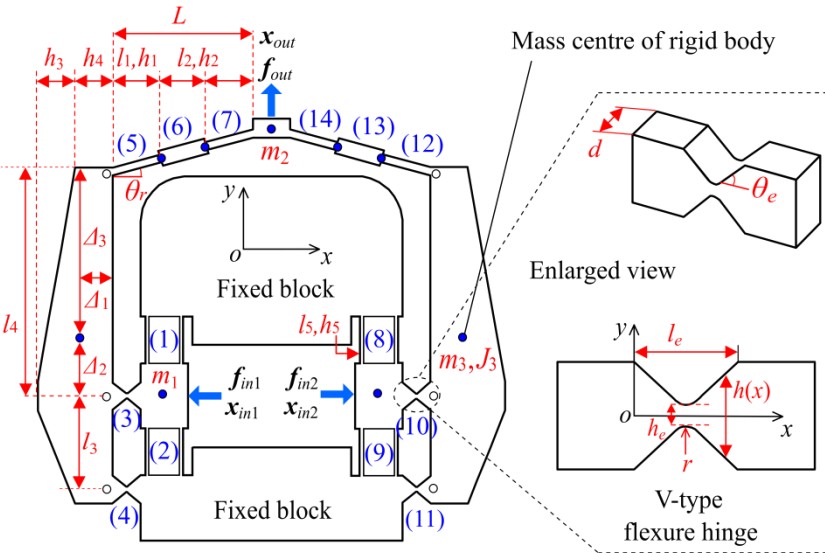

**Figure 5.** Geometric parameter, discretization, and numbering of the displacement amplification mechanism. $f_{in}$, $f_{out}$, $x_{in}$, and $x_{out}$ with the subscripts 1 and 2 denote the input/output forces and displacements. Numbers from (1) to (14) denote the elements of flexure beams/hinges. $m$ with the subscripts 1, 2, and 3 are the mass of lumped mass and rigid bodies. Other variables denote the geometric parameters of the displacement amplification mechanism.

As shown in Figure 6a, the kinetostatic and dynamic behaviors of notch-type flexure hinges/beams, can be uniformly described with the concept of the dynamic stiffness matrix in a similar form of Hook's law but is frequency dependent [38]:

$$f^e(\omega) = D^e(\omega) \cdot x^e(\omega) = \begin{bmatrix} d_1 & 0 & 0 & d_5 & 0 & 0 \\ & d_2 & -d_3 & 0 & d_6 & d_7 \\ & & d_4 & 0 & -d_7 & d_8 \\ & & & d_1 & 0 & 0 \\ & Sym & & & d_2 & d_3 \\ & & & & & d_4 \end{bmatrix} \cdot x^e(\omega) \quad (1)$$

where nodal force $f^e(\omega) = [N_j; Q_j; M_j; N_k; Q_k; M_k]$ includes the axial force, shear force and bending moment. Nodal displacement $x^e(\omega) = [u_j; w_j; \varphi_j; u_k; w_k; \varphi_k]$ includes the axial

deformation, bending deflection and rotation angle. $\omega$ is the dynamic frequency with the unit of rad/s. $d_i$ ($i = 1, 2, \ldots, 8$) are the coefficients of dynamic stiffness matrix $D^e(\omega)$, the constant values can be found in [38,39].

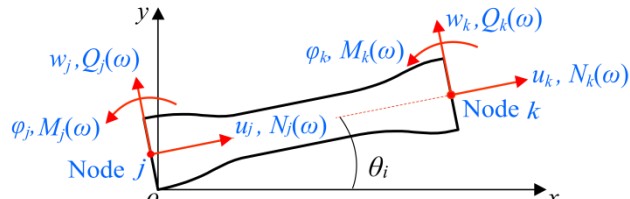

(a) Nodal force and nodal displacement of the *i*th flexure hinge

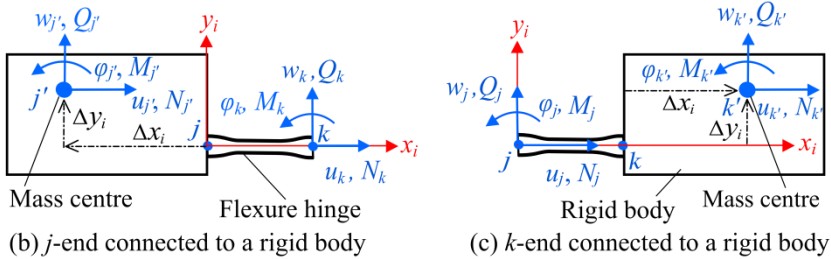

(b) *j*-end connected to a rigid body    (c) *k*-end connected to a rigid body

**Figure 6.** Nodal force and nodal displacement of flexure hinges/beams connected or not connected to a rigid body. (**a**) Nodal force and nodal displacement of the *i*th flexure hinge/beam in the reference coordinate frame *o-xy*. (**b**) The *j*-end is connected to a rigid body. (**c**) The *k*-end is connected to a rigid body.

The dynamic stiffness matrix of flexure hinges/beams in the local coordinate frame should be transformed into the reference frame *o-xy*. For flexure beams (1), (2), (6), (7), (8), (9), (13), and (14) in Figure 5, their dynamic stiffness matrices in the reference coordinate frame *o-xy* can be expressed as:

$$D_i(\omega) = \begin{bmatrix} R_i & 0 \\ 0 & R_i \end{bmatrix}^{\mathrm{T}} \cdot D^e(\omega) \cdot \begin{bmatrix} R_i & 0 \\ 0 & R_i \end{bmatrix}, \ R_i = \begin{bmatrix} \cos\theta_i & \sin\theta_i & 0 \\ -\sin\theta_i & \cos\theta_i & 0 \\ 0 & 0 & 1 \end{bmatrix} \quad (2)$$

Orientation angle $\theta_i$ of the coordinate transformation matrix $R_i$ with respect to the reference coordinate frame *o-xy* in Equation (2) are listed in Table 1. V-type flexure hinges (3), (4), (10), and (11) are connected to rigid bodies with their *k* end. As shown in Figure 6c, their dynamic stiffness matrices in the reference frame *o-xy* can be obtained by shifting the *k*-end nodes to the mass center of rigid body including the rigid motion [39]:

$$D_i(\omega) = \begin{bmatrix} R_i & 0 \\ 0 & R_{pi} \end{bmatrix}^{\mathrm{T}} \cdot D^e(\omega) \cdot \begin{bmatrix} R_i & 0 \\ 0 & R_{pi} \end{bmatrix}, \ R_{pi} = \begin{bmatrix} \cos\theta_i & \sin\theta_i & \Delta y_i \\ -\sin\theta_i & \cos\theta_i & -\Delta x_i \\ 0 & 0 & 1 \end{bmatrix} \quad (3)$$

**Table 1.** Orientation angle of flexure hinges/beams with respect to the reference coordinate frame.

| Variables | Values | Variables | Values | Variables | Values |
|-----------|--------|-----------|--------|-----------|--------|
| $\theta_1$ | $-90$ | $\theta_6$ | $\theta_r$ | $\theta_{11}$ | 0 |
| $\theta_2$ | 90 | $\theta_7$ | $\theta_r$ | $\theta_{12}$ | $180 - \theta_r$ |
| $\theta_3$ | 180 | $\theta_8$ | $-90$ | $\theta_{13}$ | $180 - \theta_r$ |
| $\theta_4$ | 180 | $\theta_9$ | 90 | $\theta_{14}$ | $180 - \theta_r$ |
| $\theta_5$ | $\theta_r$ | $\theta_{10}$ | 0 | | |

Flexure beams (5) and (12) are connected to rigid bodies with their *j* end. Their extended dynamic stiffness matrices can also be calculated in the reference frame *o-xy*:

$$D_i(\omega) = \begin{bmatrix} R_{pi} & 0 \\ 0 & R_i \end{bmatrix}^{\mathrm{T}} \cdot D^e(\omega) \cdot \begin{bmatrix} R_{pi} & 0 \\ 0 & R_i \end{bmatrix}, \; R_{pi} = \begin{bmatrix} \cos\theta_i & \sin\theta_i & \Delta y_i \\ -\sin\theta_i & \cos\theta_i & -\Delta x_i \\ 0 & 0 & 1 \end{bmatrix} \quad (4)$$

where $\Delta x_i$ and $\Delta y_i$ are the mass centre position of a rigid body with respect to the *j*-end or *k*-end of the *i*th flexure hinge/beam in the local coordinate frame $j\text{-}x_iy_i$ (their values are listed in Table 2). It is noticed a plus or minus sign for $\Delta x_i$ and $\Delta y_i$.

**Table 2.** The mass center position of a rigid body with respect to the *j*-end or *k*-end of the *i*th flexure element.

| Variables | Values | Variables | Values |
|---|---|---|---|
| $(\Delta x_3, \Delta y_3)$ | $(\Delta_1, -\Delta_2)$ | $(\Delta x_{11}, \Delta y_{11})$ | $(\Delta_1, \Delta_2 + l_3)$ |
| $(\Delta x_4, \Delta y_4)$ | $(\Delta_1, -\Delta_2 - l_3)$ | $(\Delta x_5, \Delta y_5)$ | $(-\Delta_3\sin\theta_h - \Delta_1\cos\theta_h, \Delta_1\sin\theta_h - \Delta_3\cos\theta_h)$ |
| $(\Delta x_{10}, \Delta y_{10})$ | $(\Delta_1, \Delta_2)$ | $(\Delta x_{12}, \Delta y_{12})$ | $(-\Delta_3\sin\theta_h - \Delta_1\cos\theta_h, \Delta_3\cos\theta_h - \Delta_1\sin\theta_h)$ |

Based on the above, the frequency-dependent relationship of nodal displacement and nodal force for the *i*th flexure hinge/beam in the reference coordinate frame *o-xy* can be correlated with the dynamic stiffness matrix $D_i(\omega)$ in a similar form of Hook's law:

$$\left\{ \begin{array}{c} F_{i,j} \\ F_{i,k} \end{array} \right\} = D_i(\omega) \cdot \left\{ \begin{array}{c} x_{i,j} \\ x_{i,k} \end{array} \right\} = \begin{bmatrix} k_{i,1} & k_{i,2} \\ k_{i,3} & k_{i,4} \end{bmatrix} \cdot \left\{ \begin{array}{c} x_{i,j} \\ x_{i,k} \end{array} \right\} \quad (5)$$

where $F_{i,j}(\omega) = [N_j; Q_j; M_j]$, $F_{i,k}(\omega) = [N_k; Q_k; M_k]$ and $x_{i,j}(\omega) = [u_j; w_j; \varphi_j]$, $x_{i,k}(\omega) = [u_k; w_k; \varphi_k]$ are the nodal forces and nodal displacements of the *i*th flexure hinge/beam in the reference coordinate frame.

The dynamic stiffness matrix in Equation (5) captures the relationship of nodal forces and nodal displacements of the *i*th flexure hinge/beam in a similar form of Hook's law. This relationship can be easily transformed in the form of transfer matrix:

$$\left\{ \begin{array}{c} x_{i,k} \\ -F_{i,k} \end{array} \right\} = T_i \cdot \left\{ \begin{array}{c} x_{i,j} \\ F_{i,j} \end{array} \right\} = \begin{bmatrix} t_{i,1} & t_{i,2} \\ t_{i,3} & t_{i,4} \end{bmatrix} \cdot \left\{ \begin{array}{c} x_{i,j} \\ F_{i,j} \end{array} \right\} = \begin{bmatrix} -k_{i,2}^{-1} \cdot k_{i,1} & k_{i,2}^{-1} \\ k_{i,4} \cdot k_{i,2}^{-1} \cdot k_{i,1} - k_{i,3} & -k_{i,4} \cdot k_{i,2}^{-1} \end{bmatrix} \cdot \left\{ \begin{array}{c} x_{i,j} \\ F_{i,j} \end{array} \right\} \quad (6)$$

where $T_i$ is the transfer matrix of the *i*th flexure hinge/beam in the reference frame.

With the above preparation of flexure elements, the displacement amplification mechanism in Figure 5 can be further discretized as a building block configuration, as shown in Figure 7. The detailed nodal force and nodal displacement of each building block are illustrated in the figure. The nodal displacements between two adjacent building blocks are equal, while an opposite reaction is legal for the two nodal forces of adjacent building blocks. Therefore, the total transfer matrix of the two sub-chains can be directly obtained by taking the chain path from input ports to the output port. The nodal forces and nodal displacements between the input and output building blocks can be then related with the total transfer matrices $T_I$ and $T_{II}$:

$$\left\{ \begin{array}{c} x_{7,k} \\ -F_{7,k} \end{array} \right\} = [T_I] \cdot \left\{ \begin{array}{c} x_{3,j} \\ F_{3,j} \end{array} \right\} = [T_7 \cdot T_6 \cdot T_5 \cdot T_{m3} \cdot Q_2 \cdot T_3 \cdot Q_1 \cdot T_{m1}] \cdot \left\{ \begin{array}{c} x_{3,j} \\ F_{3,j} \end{array} \right\} \quad (7)$$

$$\left\{ \begin{array}{c} x_{14,k} \\ -F_{14,k} \end{array} \right\} = [T_{II}] \cdot \left\{ \begin{array}{c} x_{10,j} \\ F_{10,j} \end{array} \right\} = [T_{14} \cdot T_{13} \cdot T_{12} \cdot T_{m3} \cdot Q_4 \cdot T_{10} \cdot Q_3 \cdot T_{m1}] \cdot \left\{ \begin{array}{c} x_{10,j} \\ F_{10,j} \end{array} \right\} \quad (8)$$

where $T_{m1}$ and $T_{m3}$ are the transfer matrix of the input port and rigid body with respect to the mass centre. $Q_i$ (*I* = 1, 2, 3, 4) are the accessional stiffness matrices of the parallel sub-chains at the *j*-end and/or *k*-end of the flexure hinges (3) and (10). Actually, $Q_i$ indicates

the force summation of parallel sub-chains at the joint point [38]. The expressions of $T_m$ and $Q_i$ are:

$$T_{mi} = \begin{bmatrix} I_3 & O_3 \\ -M_i & I_3 \end{bmatrix}, \quad M_i = -\omega^2 \cdot \begin{bmatrix} m_i & 0 & 0 \\ 0 & m_i & 0 \\ 0 & 0 & J_i \end{bmatrix} \quad (i = 1, 3) \tag{9}$$

$$\begin{cases} Q_1 = \begin{bmatrix} I_3 & O_3 \\ -2(k_{1,4} + k_{2,4}) & I_3 \end{bmatrix}, \quad Q_2 = \begin{bmatrix} I_3 & O_3 \\ -k_{4,4} & I_3 \end{bmatrix} \\ Q_3 = \begin{bmatrix} I_3 & O_3 \\ -2(k_{8,4} + k_{9,4}) & I_3 \end{bmatrix}, \quad Q_4 = \begin{bmatrix} I_3 & O_3 \\ -k_{11,4} & I_3 \end{bmatrix} \end{cases} \tag{10}$$

where $I_3$ and $O_3$ are the $3 \times 3$ unit and zero matrices. $k_{i,4}$ ($i$ = 1, 2, 4, 8, 9, 11) are the last three rows and three columns of the dynamic stiffness matrix $D_i(\omega)$ in Equation (5).

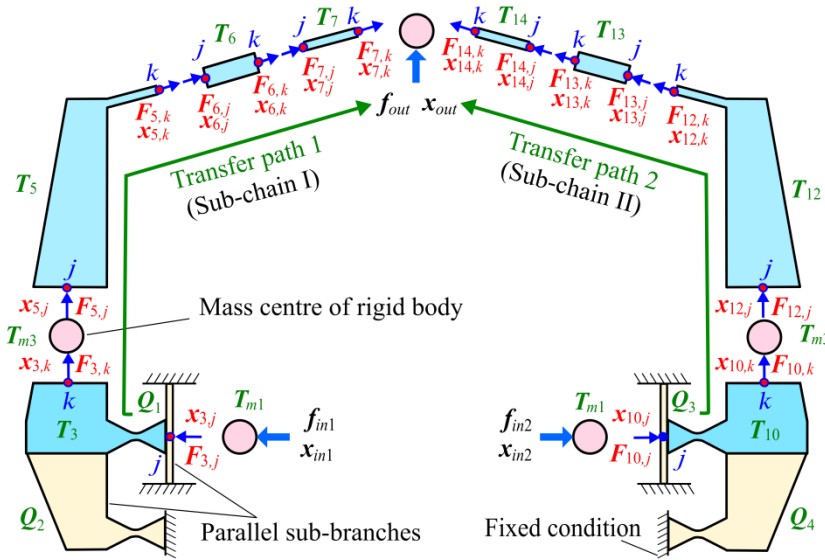

**Figure 7.** Discretized building blocks of the displacement amplification mechanism.

Based on the condensation above, the displacement amplification mechanism in Figure 7 can be simplified as an equivalent network shown in Figure 8. The relationship between the nodal force and nodal displacement of the two condensed links can be expressed again in the form of the dynamic stiffness matrix:

$$\left\{ \begin{array}{c} F_{i,j} \\ F_{i,k} \end{array} \right\} = D_i(\omega) \cdot \left\{ \begin{array}{c} x_{i,j} \\ x_{i,k} \end{array} \right\} = \begin{bmatrix} k_{i,1} & k_{i,2} \\ k_{i,3} & k_{i,4} \end{bmatrix} \cdot \left\{ \begin{array}{c} x_{i,j} \\ x_{i,k} \end{array} \right\} = \begin{bmatrix} -t_{i,2}^{-1} \cdot t_{i,1} & t_{i,2}^{-1} \\ t_{i,4} \cdot t_{i,2}^{-1} \cdot t_{i,1} - t_{i,3} & -t_{i,4} \cdot t_{i,2}^{-1} \end{bmatrix} \cdot \left\{ \begin{array}{c} x_{i,j} \\ x_{i,k} \end{array} \right\} \tag{11}$$

where $t_{i,1}$, $t_{i,2}$, $t_{i,3}$, and $t_{i,4}$ ($i$ = I, II) are the block sub-matrices of $T_I$ and $T_{II}$.

Taking the input and output ports in Figure 8 as the study objects, the following force equilibrium equation sets can be directly established based on d'Alembert's principle by summarizing the inverse nodal force, inertial force, and external force:

$$\begin{cases} f_{in1} = F_{I,j} \\ f_{in2} = F_{II,j} \\ f_{out} = F_{I,k} + F_{II,k} + M_2 \cdot x_{out} \end{cases} \tag{12}$$

where the mass matrix $M_2$ can be calculated by Equation (9) with the value of $m_2$. It is noticed that $J_2 = 0$ which will not significantly influence the dynamic performance.

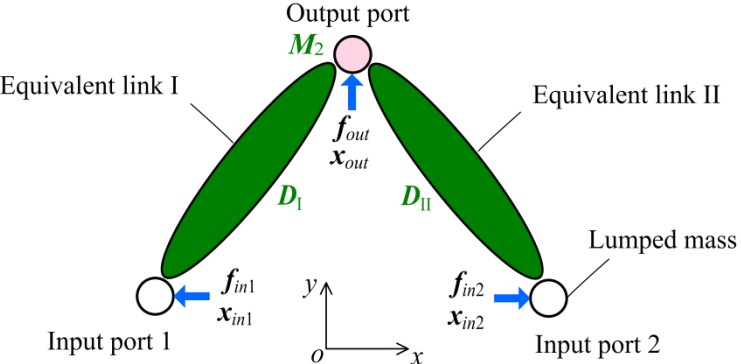

**Figure 8.** Condensed configuration of the displacement amplification mechanism.

By substituting Equation (11) into Equation (12), and re-writing the linear equation sets in the form of matrix, the two-port dynamic stiffness model of the presented displacement amplification mechanism can be ultimately derived as:

$$
\left\{ \begin{array}{c} f_{in1} \\ f_{in2} \\ f_{out} \end{array} \right\} = \boldsymbol{D}(\omega) \cdot \left\{ \begin{array}{c} x_{in1} \\ x_{in2} \\ x_{out} \end{array} \right\} = \left[ \begin{array}{ccc} \boldsymbol{k}_{I,1} & \boldsymbol{O}_3 & \boldsymbol{k}_{I,2} \\ \boldsymbol{O}_3 & \boldsymbol{k}_{II,1} & \boldsymbol{k}_{II,2} \\ \boldsymbol{k}_{I,3} & \boldsymbol{k}_{II,3} & \boldsymbol{k}_{I,4} + \boldsymbol{k}_{II,4} + \boldsymbol{M}_2 \end{array} \right] \cdot \left\{ \begin{array}{c} x_{in1} \\ x_{in2} \\ x_{out} \end{array} \right\} \qquad (13)
$$

where $\boldsymbol{k}_{I,i}$ and $\boldsymbol{k}_{II,i}$ ($i$ = 1, 2, 3, 4) are the block sub-matrices of $\boldsymbol{D}_I$ and $\boldsymbol{D}_{II}$ in Equation (11).

## 4. Parameter Influence Analysis

By solving Equation (13) with the dynamic frequency $\omega$ = 0, the static performances can be calculated as: displacement amplification ratio $R = x_{out}/2x_{in}$, input stiffness $K_{in} = f_{in}/x_{in}$, output stiffness $K_{out} = f_{out}/x_{out}$. The dynamic response spectrum can be directly obtained starting from an initial value of frequency $f$ = 1 Hz ($\omega = 2\pi f$) and incrementing step-by-step with $\Delta f$ =1 Hz. In addition, the resonance frequency in the output direction can be obtained by checking the peaks of the dynamic displacement response spectrum curve, while all the natural frequencies can be calculated by tracking the zero roots of the determinant of the overall dynamic stiffness matrix $\boldsymbol{D}(\omega)$ in Equation (13).

The static performances, the first two-order natural frequencies and the dynamic displacement response on the frequency domain were calculated by the theoretical model in Equation (13) and the finite element software package ANSYS Workbench 15.0. The geometric parameters are listed in Table 3, in which the mass $m_3$ and mass moment of inertial $J_3$ were read out from CAD software. The material was selected as aluminum alloy 7075 with the density of $\rho$ = 2770 kg/m$^3$, Young's modulus of $E$ = 71 GPa, and the shear modulus of $G$ = 27 GPa. During the finite element simulation, the static, modal, and harmonic analyses with small deformation were performed. The Solid186 element was chosen to build the model and the advanced size function of proximity and curvature was adopted to refine the elements. An input force with the magnitude of 100 N and dynamic frequencies from 1 Hz to 5 kHz was exerted.

**Table 3.** Geometric parameters of the displacement amplification mechanism.

| Parameters | Values | Parameters | Values | Parameters | Values |
|:---:|:---:|:---:|:---:|:---:|:---:|
| $l_1$ | 5.0 mm | $h_1$ | 0.6 mm | $d$ | 7.0 mm |
| $l_2$ | 5.0 mm | $h_2$ | 1.5 mm | $r$ | 0.4 mm |
| $l_3$ | 10.0 mm | $h_3$ | 4.0 mm | $L$ | 15.0 mm |
| $l_4$ | 24.0 mm | $h_4$ | 4.0 mm | $m_1$ | $\rho \times d \times 5\,\text{mm} \times 7\,\text{mm}$ |
| $l_5$ | 5.0 mm | $h_5$ | 0.4 mm | $m_2$ | $\rho \times d \times 2\,\text{mm} \times 4\,\text{mm}$ |
| $l_e$ | 3.0 mm | $h_e$ | 1.5 mm | $m_3$ | 4.2 g |
| $\theta_r$ | 8.0 deg | $\theta_e$ | 40.0 deg | $J_3$ | 0.4 kg·mm$^2$ |
| $\Delta_1$ | 3.6 mm | $\Delta_2$ | 6.5 mm | $\Delta_3$ | 17 mm |

Figure 9 provides the numerical results of the dynamic bandwidth of the amplifier in terms of the dynamic response spectrum of output displacement and zero roots of the determinant of the overall dynamic stiffness matrix $D(\omega)$. The finite element results of the first two mode shapes are also shown in Figure 9a with the first two-order natural frequencies of 1808 and 2150 Hz. The two zero roots of the determinant of the overall dynamic stiffness matrix $D(\omega)$ in Figure 9a exactly correspond to the first two-order mode shapes provided by the finite element method. A sharp peak of the lightly damped resonance emerges in the frequency response curve, as shown in Figure 9b. Assuming the finite elemental results as the benchmark, it appears that around 0.8% and 3.4% of the two resonance frequencies were overestimated by the theoretical model, which demonstrate the validity of the presented method. The errors between the theoretical model and finite element results are mainly attributed to the mismatching of geometric parameters and the rigid body assumption during the theoretical modeling. Visualized from the dynamic response spectrum in Figure 9b, the first-order spurious rocking vibration mode in the two levers was not excited by the input force. It is easy to understand that the input force only excited the movement along the output direction owing to the symmetric structure of the compliant mechanism. This indicates the fact that there is no need to design extra guiding flexure mechanisms at the output port to compensate this type of spurious vibration mode from the structural compactness point of view.

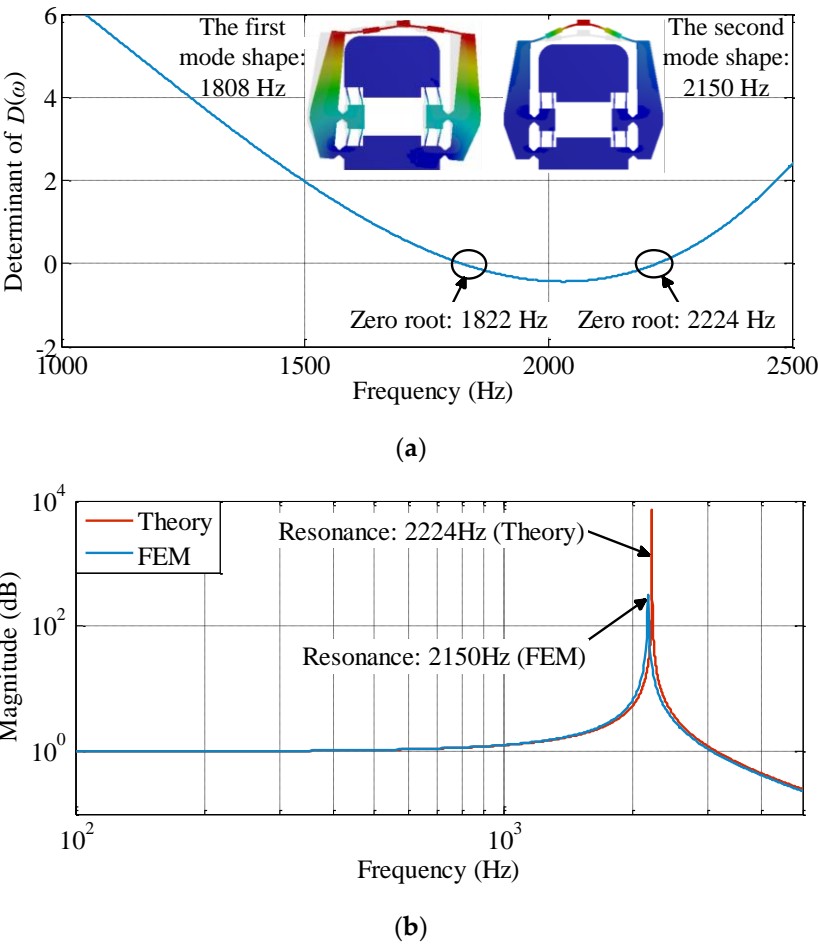

**Figure 9.** Dynamic bandwidth of the displacement amplification mechanism provided by the two-port dynamic stiffness model and finite element software package ANSYS Workbench 15.0. (**a**) Searching the zero roots of the determinant of the dynamic stiffness matrix $D(\omega)$ and the finite element results of mode shapes. (**b**) The dynamic response spectrum of the output displacement.

Table 4 shows the variation of the static performances and the first two-order natural frequencies under two sets of angle $\theta_r$. During the theoretical calculation, the lever arm was respectively regarded as rigid bodies (Theory1) and as two flexure beams (Theory2) for a comparison. The lever arm was equivalent as two flexure beams with rectangular cross-section based on the equal area, thus the in-plane thickness of the two equivalent beam elements are, respectively, 7.74 mm (corresponding to length $l_3$ in Figure 5) and 6.25 mm (corresponding to length $l_4$ in Figure 5). The modeling process for the case of Theory2 is similar to these in Section 3. It can be clearly seen from Table 4 that Theory1 and Theory2 well predicts the natural frequencies with respect to the results of FEM. However, the prediction accuracy of Theory1 by regarding the lever arm as rigid bodies is reduced for the results of the displacement amplification ratio $R$ and input stiffness $K_{in}$. This indicates the fact that the compliance of the lever arm has a significant influence on the static performances of the displacement amplification mechanism. Actually, the output stiffness $K_{out}$ and the first two natural frequencies $f_{n1}$ and $f_{n2}$ are mainly dependent on the semi bridge-type flexure amplifier, and these metrics are slightly influenced whether the lever arms are regarded as rigid bodies or flexure beams.

**Table 4.** Comparison on the static and dynamic performances of the displacement amplification mechanism provided by the presented model and finite element method (FEM).

| Angle | Methods | Static Performances | | | The First Two-Order Natural Frequencies | |
|---|---|---|---|---|---|---|
| | | $R$ | $K_{in}$ (N/µm) | $K_{out}$ (N/µm) | $f_{n1}$ (Hz) | $f_{n2}$ (Hz) |
| $\theta_r$ = 8 deg | Theory1 | 10.54 | 20.27 | 0.078 | 1822 | 2224 |
| | Theory2 | 5.39 | 11.23 | 0.077 | 1790 | 2223 |
| | FEM | 5.84 | 10.45 | 0.072 | 1808 | 2150 |
| | Error1 | 80.48% | 93.97% | 8.33% | 0.77% | 3.44% |
| | Error2 | 7.71% | 7.46% | 6.94% | 1.00% | 3.40% |
| $\theta_r$ = 20 deg | Theory1 | 4.62 | 5.92 | 0.132 | 1803 | 2089 |
| | Theory2 | 3.88 | 5.28 | 0.126 | 1777 | 2094 |
| | FEM | 3.91 | 4.82 | 0.123 | 1794 | 2073 |
| | Error1 | 18.20% | 22.82% | 7.31% | 0.50% | 0.77% |
| | Error2 | 0.77% | 9.54% | 2.43% | 0.95% | 1.01% |

Figures 10–12 theoretically show the displacement amplification ratio considering the axial stiffness of piezoelectric stacks [$R \cdot K_p / (K_{in} + K_p)$, here $K_p$ = 100 N/µm] and the main resonance frequency (i.e., the second-order natural frequency) of the mechanism versus three key geometric parameters: (a) thickness $h_5$ of the guiding flexure beams, (b) length $l_2$ of the semi bridge-type amplifier, and (c) minimum thickness $h_e$ of the V-type flexure hinges. The displacement amplification ratio reduces with the increase of thickness $h_5$ and length $l_2$, but it is nonsensitive at the domain of small values of $l_2$. Actually, the compliance of flexure arm of the semi bridge-type compliant amplifier is mainly dependent on the in-plane thickness $h_1$ when the length $l_2$ is small. Therefore, displacement amplification ratio is slightly influenced by small values of $l_2$. A suitable value of 0.4 mm can be confirmed for $h_5$, and the length $l_2$ can be selected as 5 mm for the prototype. In addition, the displacement amplification ratio increases with a larger minimum thickness $h_e$ of the V-type flexure hinges and reaches to a maximum value due to the attenuation effect of input stiffness. The dynamic resonance frequency continuously increases with the increase of $h_e$, and a relatively large thickness $h_e$ of 1.5 mm can be optimally selected for fabricating the prototype.

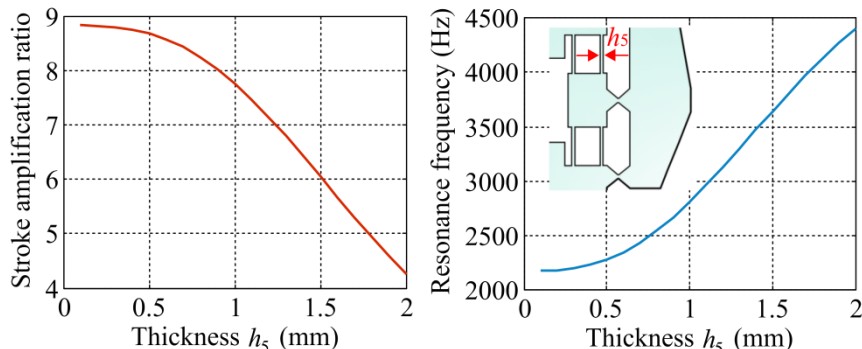

**Figure 10.** Theoretical results of the displacement amplification ratio and the main resonance frequency for the displacement amplification mechanism versus the thickness of guiding flexure beams.

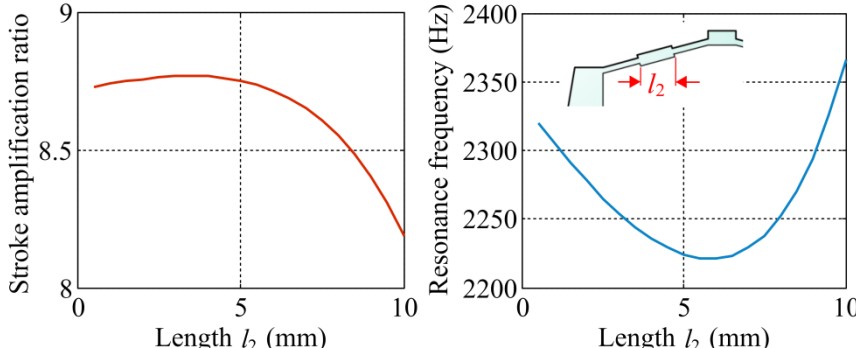

**Figure 11.** Theoretical results of the displacement amplification ratio and the main resonance frequency for the displacement amplification mechanism versus the length of internal links of the semi bridge-type amplifier.

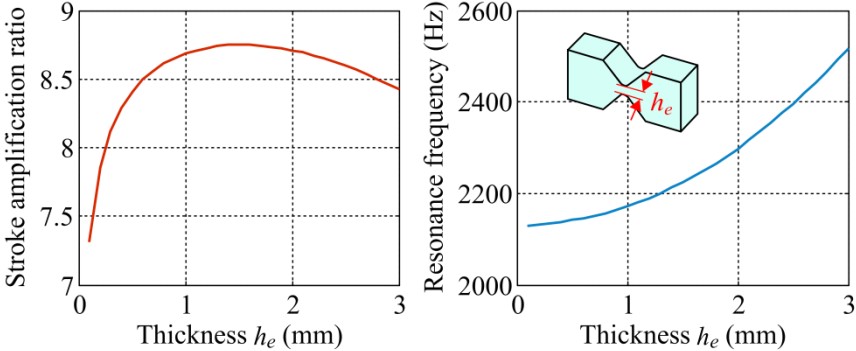

**Figure 12.** Theoretical results of the displacement amplification ratio and the main resonance frequency for the displacement amplification mechanism versus the minimum thickness of V-type flexure hinges.

## 5. Prototype and Experimental Testing

A prototype of the proposed amplified piezoelectric actuator was fabricated and experimentally tested, as shown in Figure 13. The material of the displacement amplification mechanism is aluminum alloy 7075 with the density of $\rho$ = 2770 kg/m$^3$ and Young's modulus of $E$ = 71 GPa. The amplifier was fabricated through the wire-electrode cutting technique. Two through-holes on the input ports were reserved to exert a tensile force in order to increase the distance between the two input ports for assembling piezoelectric stacks into the displacement amplification mechanism by the interference fit with a tolerance of about 20 μm. The whole amplifier is sized as 50 mm × 44 mm × 7 mm, and it was mounted on an optical table for the static and dynamic measurements with the

reduced ground vibration. Piezo-stacks (18 mm × 7 mm × 7 mm) was excited using a power amplifier (PI Corp., E-500). The used piezoelectric stacks have the axial stiffness of 100 N/μm and the output stroke of about 16 μm under the input voltage of 100 V. The output displacement of the amplified piezoelectric actuator was measured using a precision laser sensor (Keyence, LK-G10) and recorded by a data recorder (Yiheng Inc., Hangzhou, China). Non-negative sinusoidal signals at the working frequency of 1 Hz was used for the static displacement and hysteresis evaluation. The dynamic responses were measured by sweeping the frequencies from 1 to 4000 Hz and step signals, respectively. All of the tests were implemented at room temperature.

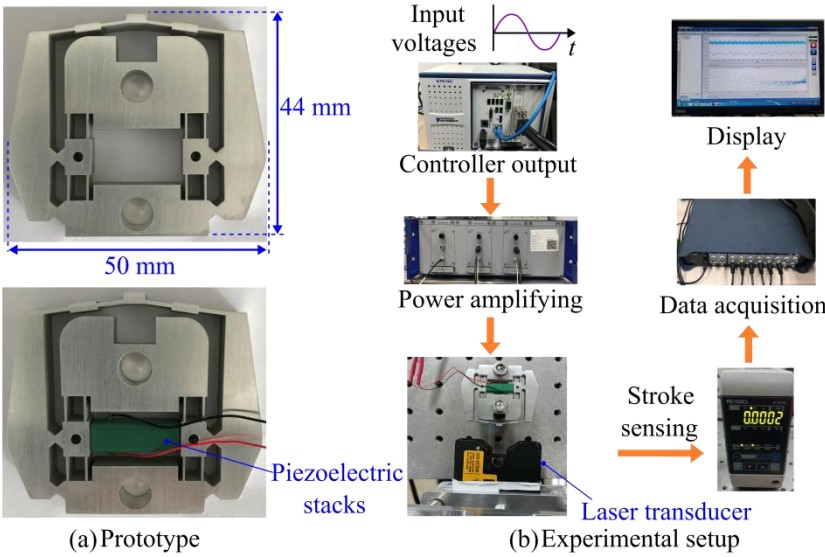

**Figure 13.** Prototype and the experimental setup for measuring the static and dynamic performances.

The static output displacement at the working frequency of 1 Hz and input voltage of 100 V is shown in Figure 14, from which steady and smooth waves can be observed. The maximum input and output static displacements can be read out as 10 and 61 μm with the displacement amplification ratio of 6.1. In addition, the hysteresis error of around 17% from piezoelectric material can be observed and this type of error can be further compensated by advanced control strategies that is not the emphasis of the current study. The parasitic motion error of the output port along the $x$ direction was also measured, as shown in Figure 14c. The maximum parasitic motion error is about 0.25 μm under the output displacement of 61 μm, which corresponds to the relative error of 0.41%. The small parasitic motion error is attributed to the symmetric and compact structure of the design.

The dynamic response of the prototype was recorded by sweeping the frequency, and the results on both time and frequency domain are shown in Figure 15. The single peak in the response curve corresponds to the vibration mode in the output direction, and the measuring result is 2176 Hz. From the dynamic response curve in Figure 15b, the first-order local spurious swing and rocking vibration mode in Figure 9a was indeed not excited and only the main vibration mode along the output direction appear. The experimental result of single resonance peek further indicates that there is no necessity to design guiding flexure beams at the output port from the compactness point of view.

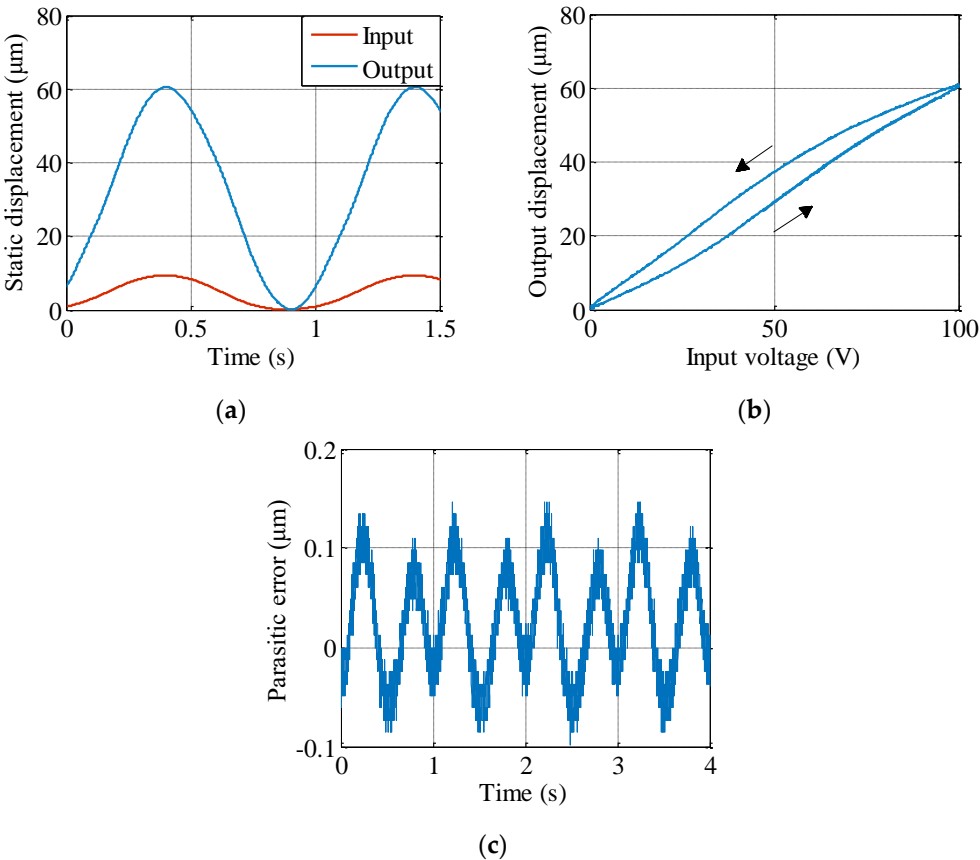

**Figure 14.** Experimental results of the static input and output displacements of the presented displacement amplification mechanism. (**a**) Input and output displacements. (**b**) Open-loop hysteresis characteristic. (**c**) Parasitic motion error of the output port in the *x* direction.

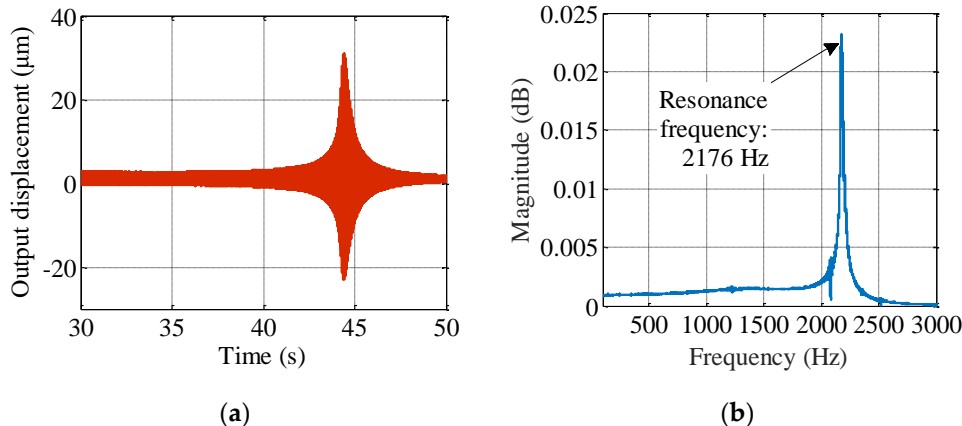

**Figure 15.** Experimental results of the dynamic response of the presented displacement amplification mechanism. (**a**) Output on the time domain. (**b**) Response on the frequency domain.

     The third experiment was performed to obtain the step response time in an open-loop control. In this case, two step transient signals with the magnitude of 100 V were followed by the amplified piezoelectric actuator. From the results in Figure 16, it can be seen that the step response time obtained necessary to reach 90% of the targeted stroke is about 0.4 ms. The fast step response time in a matter of sub-microseconds is superior which is attributed to the refined design in the current study.

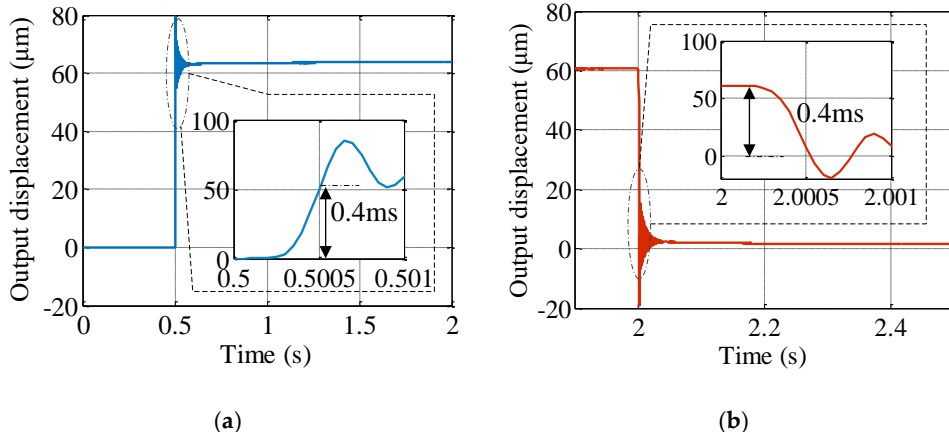

**Figure 16.** Experimental results of the step response for the presented displacement amplification mechanism. (**a**) Step increasing. (**b**) Step reducing.

To summarize, Table 5 compares the maximum output displacement and the main resonance frequency of the presented amplified piezoelectric actuator with other typical ones in literature. The findings show that very compact structure can be achieved in some previous studies with satisfying output displacements, such as the design in [35]. The current design has a little larger size in comparison to this one, but the presented amplified piezoelectric actuator exhibits much higher dynamic bandwidth. In addition, some reported compliant amplifying mechanisms in literature have a relatively large displacement amplification ratio, but the dynamic resonance frequency and the compactness still needs to be further improved in comparison to the current design. The tradeoff among the displacement amplification ratio, dynamic bandwidth and structural compactness is still a challenging issue in the field of compliant mechanisms. The current study attempts to equilibrate the tradeoff by combining the lever-type and semi bridge-type compliant amplifying mechanisms in a compact configuration. The optimal structural parameters were also confirmed by the parameter influence analyses with a comprehensive two-port dynamic stiffness model. Further investigations on inventing better configurations and some advanced closed-loop controller are deserved, which are expected for potential applications requiring small space, high motion speed and high accuracy. These potential application scenarios would involve precision positioning stages, micro-grippers, fast mechanical switches and clutches, as well as jet dispensers, to name a few.

**Table 5.** Comparison on the static and dynamic performances of different amplified piezoelectric actuators.

| Amplifiers | Size: Length × Width × Height | Output Stroke (Displacement Amplifying Ratio) | Resonance Frequency |
|---|---|---|---|
| Ref. [7] | 120 mm × 80 mm × 25 mm (Approximate value) | 30 μm ($R$ = 10.4) | 1152 Hz |
| Ref. [35] | 30 mm × 30 mm × 15 mm | 80 μm ($R$ = 10) | 190 Hz |
| Ref. [31] | 98 mm × 52 mm × 20 mm | 69 μm ($R$ = 3.51) | 457 Hz |
| Ref. [31] (Traditional bridge-type) | 98 mm × 52 mm × 20 mm | 74 μm ($R$ = 3.70) | 355 Hz |
| Ref. [36] | 134 mm × 50 mm × 20 mm (Approximate value) | 200 μm ($R$ = 20) | 189 Hz |
| Ref. [40] | 92 mm × 50 mm × 18 mm | 214 μm ($R$ = 12.1) | 205 Hz |
| Ref. [41] | 65 mm × 22 mm × 10 mm | 200 μm ($R$ = 16.2) | 628 Hz |
| The presented | 50 mm × 44 mm × 7 mm | 61 μm ($R$ = 6) | 2176 Hz |

## 6. Conclusions

In this paper, we focused on exploring the benefits of synthesizing different types of compliant mechanisms to mechanically amplify the micro displacement of piezoelectric stacks. The design purpose is to achieve a large displacement amplification efficiency while guaranteeing a high dynamic bandwidth within a compact size. The following conclusions can be reached:

(1) Combining lever-type and semi bridge-type complaint amplifying mechanisms, we proposed a new amplified piezoelectric actuator. Although these two types of compliant mechanisms might not be perfect from a practical point of view, our combination enables an improved performance. We showcase by comparing with previous designs such a hybrid displacement amplification mechanism can reach a resonance frequency of 2.1 kHz and the displacement amplification ratio of 6 within a compact size of 50 mm × 44 mm × 7 mm.

(2) The benefits of using a comprehensive two-port dynamic stiffness model for the dimension synthesis have been validated by extensive studies of the parameter influence analysis. The traditional transfer matrix method, which is rarely used in the presence of serial-parallel compliant mechanisms including rigid bodies and complex branches, now is able to successfully utilize for such a complicated application scenario in a way of conciseness.

It is noticed that the static and dynamic behaviors of piezoelectric stacks, such as the hysteresis and creep as well as coupling between the piezoelectric stacks and compliant amplifying mechanism, were not involved in the current study. The future work will focus on developing advanced control strategies for precision tracking control and compensating the nonlinear hysteresis and creep errors of the presented amplified piezoelectric actuator.

**Author Contributions:** Conceptualization, M.L. and X.Z.; methodology, M.L.; software, L.Y.; validation, L.Y.; experiment, T.H. and Z.L.; data curation, M.L.; writing, M.L.; supervision, X.Z.; project administration, M.L.; funding acquisition, M.L. All authors have read and agreed to the published version of the manuscript.

**Funding:** This research was funded by the National Natural Science Foundation of China [grant number 52075179], and the National Defense Technology Foundation Program of China [grant number JSHS2018212C001].

**Institutional Review Board Statement:** Not applicable.

**Informed Consent Statement:** Not applicable.

**Data Availability Statement:** Data are available upon email the corresponding authors.

**Conflicts of Interest:** The authors declare no conflict of interest.

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
