# Peer review of "Enhancing Dynamic Bandwidth of Amplified Piezoelectric Actuators by a Hybrid Lever and Bridge-Type Compliant Mechanism"

_actuators, doi:10.3390/act11050134_

Round 1

Reviewer 1 Report

In my opinion, the manuscript is suitable for publication in Actuators journal but Authors must complete a major revision. Manuscript should be revised according to following comments:

1. Chapter “Operational principle and configuration” must be improved:
a) The text on lines 98 - 104 does not refer to the subject of the article. This is part of the template,
b) Figure 2 should be supplemented with a piezoelectric stack,
c) The authors should describe the methodology of numerical simulations, the results of which are presented in Fig. 4.

2. Chapter “Parametric formulation” must be corrected:
a)    The meaning of the symbols used in Fig. 5 should be explained below the figure,
b) For equations from (1) to (13) authors should provide literature source,
c) The force generated by the piezoelectric stack has generally been called the external force. The math model of the stack must be introduced. This model should take into account the phenomenon of hysteresis and creep.

3. Chapter “Parameter influence analysis” must be corrected:
a) numerical research should take into account the properties and operation of the piezoelectric stack.

4. Chapter “Prototype and experimental testing” must be improved:
a) A block diagram of a laboratory stand should be added,
b) The properties of used Piezo-stacks should be shown,
c) Why was this stack used?
d) The authors should describe in detail how the piezoelectric stack is connected to the mechanical structure.
e) The parameters of the measuring instruments used should be given. This will allow to assess the correctness of the research methodology,
f) Fig. 16 shows the spikes for a stepwise increase in the voltage controlling the piezoelectric stack. The test results for step reduction of this voltage should also be presented.

5. Chapter “Conclusions” must be improved:
a) The conclusions should be extended to include the influence of the piezoelectric stack properties on the actuator performance.

General comment: The authors focused on the operation of the mechanical structure of the actuator without the analysing of piezoelectric stack. Taking into account that the title of the article includes "piezoelectric actuator", Authors should focus on the analysis of piezoelectric stack operation in their proposed mechanical structure in revised version of article.

Author Response

We thank the reviewer fro his/her time on reviewing our manuscript. 

Reviewer 2 Report

This paper presents a two-stage displacement flexure amplifier combining the lever-type and semi bridge-type compliant mechanisms. Moreover, a comprehensive two-port dynamic stiffness model to predict the static and dynamic behaviors of the amplifier. This paper can be accepted for publication after some revisions.

  1. In section 4, finite element results find the existence of the two mode shapes with the natural frequencies of 1798 Hz and 2150 Hz. The theoretical model also predicts these two frequencies and the deviation of 1.3% and 3.4% are also obtained. However, it is confused that authors deny the existence of the first-order local spurious swing and rocking vibration mode with no explanation in the experimental part, which reduces the reliability of the finite element and the theoretical model.
  2. Authors claims that the key features of this design include minimized parasitic motion errors due to the monolithic and symmetric in-plane structure, which is also not proved experimentally.
  3. The output displacement is marked in a wrong direction in Figure 2.
  4. The clarity of these figures shall be improved.
  5. Guidelines of this journal still remain in the beginning of Section 2 and Section 5.

Author Response

(The authors gave the same response as above.)

Reviewer 3 Report

  1. The manuscript proposed a hybrid lever and bridge-type compliant mechanism to enhance dynamic bandwidth of amplified piezoelectric actuators.
  2. Page 3 lines 98 ~ 104, are unrelative to the manuscript, should be deleted.
  3. Page 11 lines 337 ~ 340, are unrelative to the manuscript, should be deleted.
  4. The resonance frequency was verified by the finite element method as shown in Figure 9(b). It is necessary to validate the output displacement too
  5. Page 10 line 316 ~ 318, the displacement amplification ratio reduces with the increase of thickness h5 and length l2, but it is nonsensitive at the domain of small values of h5 and l2. Can the authors explain the statement ?

Author Response

(The authors gave the same response as above.)

Round 2

Reviewer 1 Report

I accept in present form.

Reviewer 3 Report

The manuscript has been revised appropriately.